# Imipramine Accelerates Nonalcoholic Fatty Liver Disease, Renal Impairment, Diabetic Retinopathy, Insulin Resistance, and Urinary Chromium Loss in Obese Mice

**DOI:** 10.3390/vetsci8090189

**Published:** 2021-09-09

**Authors:** Geng-Ruei Chang, Po-Hsun Hou, Chao-Min Wang, Jen-Wei Lin, Wei-Li Lin, Tzu-Chun Lin, Huei-Jyuan Liao, Chee-Hong Chan, Yu-Chen Wang

**Affiliations:** 1Department of Veterinary Medicine, National Chiayi University, 580 Xinmin Road, Chiayi 60054, Taiwan; grchang@mail.ncyu.edu.tw (G.-R.C.); leowang@mail.ncyu.edu.tw (C.-M.W.); lin890090@gmail.com (T.-C.L.); pipi324615@gmail.com (H.-J.L.); 2Department of Psychiatry, Taichung Veterans General Hospital, 4 Section, 1650 Taiwan Boulevard, Taichung 40705, Taiwan; peterhopo2@yahoo.com.tw; 3Faculty of Medicine, National Yang-Ming University, 2 Section, 155 Linong Street, Beitou District, Taipei 11221, Taiwan; 4College of Medicine, National Chung Hsing University, 145 Xingda Road, South District, Taichung 40227, Taiwan; 5Bachelor Degree Program in Animal Healthcare, Hungkuang University, 6 Section, 1018 Taiwan Boulevard, Shalu District, Taichung 433304, Taiwan; jenweilin@hk.edu.tw (J.-W.L.); ivorylily99@gmail.com (W.-L.L.); 6General Education Center, Chaoyang University of Technology, 168 Jifeng Eastern Road, Taichung 413310, Taiwan; 7Division of Nephrology, Chang Bing Show Chwan Memorial Hospital, 6 Lugong Road, Lukang Township, Changhua 50544, Taiwan; 8Division of Cardiology, Asia University Hospital, 222 Fuxin Road, Wufeng District, Taichung 41354, Taiwan; 9Department of Medical Laboratory Science and Biotechnology, Asia University, 500 Lioufeng Road, Wufeng District, Taichung 41354, Taiwan; 10Division of Cardiovascular Medicine, China Medical University Hospital, 2 Yude Road, North District, Taichung 404332, Taiwan; 11College of Medicine, China Medical University, 91 Hsueh-Shih Road, North District, Taichung 404333, Taiwan

**Keywords:** fatty liver disease, glucose, imipramine, insulin, obesity, renal impairment, retinal injury

## Abstract

Imipramine is a tricyclic antidepressant that has been approved for treating depression and anxiety in patients and animals and that has relatively mild side effects. However, the mechanisms of imipramine-associated disruption to metabolism and negative hepatic, renal, and retinal effects are not well defined. In this study, we evaluated C57BL6/J mice subjected to a high-fat diet (HFD) to study imipramine’s influences on obesity, fatty liver scores, glucose homeostasis, hepatic damage, distribution of chromium, and retinal/renal impairments. Obese mice receiving imipramine treatment had higher body, epididymal fat pad, and liver weights; higher serum triglyceride, aspartate and alanine aminotransferase, creatinine, blood urea nitrogen, renal antioxidant enzyme, and hepatic triglyceride levels; higher daily food efficiency; and higher expression levels of a marker of fatty acid regulation in the liver compared with the controls also fed an HFD. Furthermore, the obese mice that received imipramine treatment exhibited insulin resistance, worse glucose intolerance, decreased glucose transporter 4 expression and Akt phosphorylation levels, and increased chromium loss through urine. In addition, the treatment group exhibited considerably greater liver damage and higher fatty liver scores, paralleling the increases in patatin-like phospholipid domain containing protein 3 and the mRNA levels of sterol regulatory element-binding protein 1 and fatty acid-binding protein 4. Retinal injury worsened in imipramine-treated mice; decreases in retinal cell layer organization and retinal thickness and increases in nuclear factor κB and inducible nitric oxide synthase levels were observed. We conclude that administration of imipramine may result in the exacerbation of nonalcoholic fatty liver disease, diabetes, diabetic retinopathy, and kidney injury.

## 1. Introduction

Imipramine, derived from dibenzazepine, is a prototypical tricyclic antidepressant (TCA). TCAs have a structure that is close to that of a phenothiazine, and they contain a tricyclic ring system. The central ring contains an alkylamine substituent [1]. On the basis of animal studies, imipramine is known as a candidate therapy for antipredator defensive behavior, sleep deprivation, and related anxiety- and depressive-like behaviors [2,3,4]. In individuals without depression, imipramine has no impact on arousal or mood; however, it may act as a sedative. In people with depression, their mood is positively affected by imipramine use [5]. Thus, it is among the most effective drugs for the treatment of severe long-term depression [6]. TCAs strongly inhibit the reuptake of norepinephrine and serotonin. 3° TCAs (i.e., tertiary TCAs), including amitriptyline and imipramine, have stronger inhibition of serotonin reabsorption than 2° TCAs, such as desipramine and nortriptyline [7]. The particular mechanism explaining the treatment-related benefits of imipramine is still not well understood. TCAs are able to blockade muscarinic, histamine H1, and α1-adrenergic receptors, which explains their anticholinergic, sedative, and hypotensive impacts, respectively [8]. The anticholinergic and sedative actions of imipramine are less pronounced than those of other 3° TCAs such as clomipramine and amitriptyline. In children, imipramine is used as an antidepressant and as a treatment for bedwetting. Off-label uses include the (1) treatment of panic disorders without or with agoraphobia, (2) second-line treatment of attention deficit hyperactivity disorder in young people, (3) management of bulimia, (4) treatment of postacute and posttraumatic stress, and (5) short-term treatment of acute depression in patients with schizophrenia and bipolar disorder [9,10,11,12]. Side effects such as drowsiness, dry mouth, excitement, and weight or appetite alterations may be induced by imipramine [13]. In-depth explorations of the side effects of imipramine have been inconclusive.

Obesity is a major risk factor for various conditions related to metabolic syndrome, such as hyperlipidemia, nonalcoholic fatty liver disease (NAFLD), insulin resistance (IR), and type 2 diabetes [14]. An imbalance between expenditure and consumption causes an increase in energy storage within the body, which could lead to weight gain and obesity [15]. Moreover, recent studies have suggested that atypical glucose–insulin homeostasis is linked to various depression severity indicators and could be a neurohormonal mediator of certain depression symptoms, such as neurocognitive impairments, and disorders that are comorbid with depression, such as cardiovascular disease [4,5,16]. Gupta et al. reported a significant association between imipramine administration and increased blood glucose levels of rabbits. They also indicated that hyperglycemic responses increased more when adrenaline and imipramine were administered simultaneously than when they were administered alone [17]. In addition, some scholars have observed the effects that the tricyclic antidepressants trimipramine and amitriptyline have on glucose homoeostasis in rats, indicating that their long-term administration could cause IR and diabetes [18]. Obesity is regularly linked to NAFLD, a group of conditions that can take the form of cirrhosis, nonalcoholic steatohepatitis, and fatty liver [19]. As well as drug-related factors including liver metabolism, daily dose, and chemical structure, different risk factors heighten the likelihood of liver injury induced by drugs [20]. How patients with fatty liver further develop nonalcoholic steatohepatitis is complicated; the mechanism could involve various factors such as the overproduction of reactive oxygen species, lowered reactive oxygen species (ROS) detoxification, and heightened profibrogenic and proinflammatory cytokine release [21]. In addition, tricyclic antidepressants may have adverse health effects associated with kidney damage, diabetes, and FLD, and their prolonged use may impede weight control and aggravate diabetes [22].

Additionally, obesity is a major renal disease risk factor. Similar to hypertension and diabetes, it heightens the risk of end-stage renal disease (ESRD) and chronic kidney disease (CKD) [23]. Moreover, in ESRD or CKD, associations exist between depression and unfavorable quality of life as well as poor health outcomes [24]. One study highlighted that symptoms of depression could be related to reduced kidney function and dialysis commencement; such symptoms should be monitored in all disease stages in CKD patients [25]. Long-term hyperglycemia and IR are among the key factors related to blood–retina barrier dysfunction and retina damage linked to diabetes [26]. The retina is crucial to human vision, and diabetes is, of course, a risk factor for diabetic retinopathy (DR). Heightened inflammation and oxidative stress are suggested to be the major mechanisms behind neural retina damage in diabetes [27,28]. Research has indicated that within 3 years of antipsychotic medication initiation in patients with schizophrenia who have visual disturbances, visual cortex function and retinal thickness gradually deteriorate [29]. Other studies have indicated that processes including vasorelaxation through alpha adrenergic blocking or direct impacts on the retinal vascular endothelium could be the reasons for cystoid macular edema induced by antidepressant use [30]. Thus, we sought to understand whether imipramine affects the retina and to elucidate the possible mechanisms. 

The adverse effects of imipramine likely arise from differences between the conditions under metabolic syndrome that affect body weight and hyperglycemia associated with imipramine. Thus, in our investigation, mice were fed a high-fat diet (HFD) and administered imipramine to mimic obesity; this was conducted to investigate imipramine’s effects on patients with obesity concomitant with psychotic disorders. The majority of research on the metabolism-related side effects of imipramine has centered on the central nervous system [4,31,32]. We examined glucose levels, lipid metabolism, and oxidative stress, and then we explored the effects of these on the kidneys caused by abnormal metabolism, particularly related to chromium; this topic is little understood. Chromium is involved in normal lipid, protein, and carbohydrate metabolism and can benefit individuals with diabetes, glucose intolerance, obesity, or nephropathy [22]. These results provide a deeper insight into metabolic impacts and the mechanisms of fatty liver as well as kidney damage stemming from chronic imipramine use as an antipsychotic in humans or animals. We also conducted investigations on whether imipramine exacerbated metabolic abnormalities, chromium changes, oxidative stress, liver and kidney damage, and DR. 

## 2. Materials and Methods

### 2.1. Animals with Feed-Induced Obesity Administered Imipramine

Five-week-old male C57BL/6J mice were procured from Taiwan’s National Laboratory Animal Center. Their treatment accorded with the Taiwanese government’s Guidelines for the Care and Use of Laboratory Animals. The protocol for using the experimental mice was approved after a review by the institutional animal care and use committee of the first author’s university (approval no. 109019). For 12 weeks, the mice were continuously administered an HFD from PMI Nutrition International (St. Louis, MO, USA; diet 58Y1, comprising 23.6% protein, 60.0% of energy from fat, 2.6 µg/g Cr, metabolizable energy: 5.16 kcal/g). To induce obesity, the HFD was administered for 12 weeks (in the literature, 4 weeks was the associated diet duration) [33]. We then categorized the group into two subgroups. During the 8-week treatment, while remaining on the HFD, one group was administered 10 mg/kg of oral imipramine (Sigma-Aldrich, St. Louis, MO, USA) through daily gavage, and the other was administered oral saline. At the start of the treatment, the control mice weighed 28.9 ± 1.48 g, and those receiving imipramine weighed 28.35 ± 1.09 g (*p* > 0.05). The imipramine dose was decided after consultation of the literature on imipramine administration in mouse model studies related to diabetes-like status, neurological behavior observations, antidepressant effects, stress, and memory deficits [34,35,36,37]. In addition, obese mice were administered oral imipramine (5 mg/kg/day) in our preliminary investigation; however, differences in weight gain and body weight were not significant between the imipramine-treated and control mice (Appendix A). Accordingly, we adopted 10 mg/kg as the imipramine dose in this study. Mice were individually housed in micro-isolation cages on HEPA-filtered and ventilated racks. The humidity (55% ± 5%) and temperature (22 ± 1 °C) were controlled, and the mice were maintained in a 12 h light/dark cycle. The mice freely accessed water and food. From the commencement of the experiment and on a weekly basis, food consumed and body weight were recorded. When the experiment was terminated, the mice were anesthetized for tissue and blood serum harvesting. Imipramine’s impact on food intake, body weight, adipocyte concentration, levels of blood glucose, fatty liver, biochemical alterations, hepatic triglycerides, endocrine profiles, insulin signaling, and renal pathology was investigated. In addition, for urine sample collection, mice were kept for 12 h in individual metabolic cages (SN-783-0; AS ONE, Osaka, Japan) before they were killed.

### 2.2. Measurement of Food Intake, Body Weight, and Leptin and Insulin Levels

Every week of the study, we measured food consumed and body weight. To determine food intake, food that remained within each cage dispenser was weighed, as was the food that remained on the cage floor. Furthermore, after tissue and blood samples were harvested, serum leptin and insulin levels were measured following a 12 h fast using an ELISA mouse insulin kit (INSKR020; Crystal Chem, Downers Grove, IL, USA) and a leptin kit (INSKR020; Crystal Chem), respectively.

### 2.3. Measurement of Serum and Hepatic Triglycerides, Creatinine, Alanine Aminotransferase, Blood Urea Nitrogen, Aspartate Aminotransferase, Serotonin, Soluble Leptin Receptor, and Fibroblast Growth Factor-21

Serum triglyceride, alanine aminotransferase (ALT), aspartate aminotransferase (AST), blood urea nitrogen (BUN), and creatinine levels were determined from the collected blood samples by means of an automated analyzer (Catalyst One Chemistry Analyzer, IDEXX Laboratories, Westbrook, ME, USA) and commercial kits under the guidance of manufacturer-approved protocols; the variation coefficient between and within analyses was under 2%. Following Folch et al., we extracted hepatic triglycerides in a 2:1 (vol/vol) mixture of chloroform and methanol [38]. Subsequently, extract solubilization was performed; the extracts were twice heated gradually to 90 °C over 5 min and subsequently cooled to room temperature. Insoluble materials were removed through centrifugation. Finally, colorimetric assay-based triglyceride analysis was conducted using the supernatant and a BioVision triglyceride quantification kit (Milpitas, CA, USA). Mouse enzyme-linked immunosorbent assay kits were employed to measure serotonin, serum soluble leptin receptor, and fibroblast growth factor-21 (FGF21) levels (EL-M0543, EL-M0545, and EL-M0435, respectively; Zgenebio Biotech, Taipei, Taiwan).

### 2.4. Histological and Morphometric Analysis

We weighed the retroperitoneal and epididymal fat pads and kidneys, spleen, liver, and heart; the weights are presented as a proportion (%) of body weight. Hematoxylin and eosin staining (BioTnA, Kaohsiung, Taiwan) was used to reveal hepatic fat infiltration, with scores of 0, 1, 2, 3, and 4 indicating no visible fat, <5% fat infiltration on the liver surface, 5% to 25% infiltration of fat, 25% to 50% fat infiltration, and >50% infiltration of fat, respectively [14,39]. Moreover, many epididymal white adipose tissue (EWAT) and retroperitoneal WAT (RWAT) sections were collected and analyzed in terms of adipocyte number and size. Hematoxylin and eosin staining of sections was conducted. We analyzed ≥10 fields (approximately 100 adipocytes) per slide for every sample [40,41]. Image acquisition was conducted by means of a high-resolution digital microscope (Moticam 2300, Motic Instruments, Canada), and adipocyte size analysis was conducted using Motic Images Plus 2.0. Correlations between the sizes of adipocytes and their distributions (%) were conducted for HFD controls and HFD mice administered imipramine. In addition, the right eyeball was fixed in a solution containing 4% paraformaldehyde, followed by dehydration, which involved passing the eyeball over several graded concentrations of ethanol (70%, 80%, 95%, and 100%). To render them transparent, the eyeballs were dehydrated and placed in xylene. Subsequent to being embedded in paraffin, mouse eyeballs were sliced into sections for hematoxylin and eosin stains [42].

### 2.5. Intraperitoneal Glucose Tolerance Test (IPGTT)

After 7 weeks of the imipramine or saline protocol, we performed an IPGTT on mice with an obesity-like status that were starved overnight but had ad libitum water. The concentration used for the assay was 1 g of glucose per 1 kg of body weight; in animal obesity and diabetes models, this is appropriate for examining antidiabetes activity [22,43]. Blood glucose levels were determined 0, 30, 60, 90, and 120 min after intraperitoneal glucose was administered. We extracted tail vein blood using a One Touch glucose meter (LifeScan, Malpitas, CA, USA). Over the 0–120 min after administration of glucose, we conducted glucose tolerance tests based on area under the curve (AUC) values.

### 2.6. Insulin Sensitivity (IS) and IR Indexes

Fasting blood glucose is widely utilized to determine IR and IS indexes [15,33,39]. We therefore used these indexes for assessment of IR and β-cell secretion function following imipramine administration. The homeostasis model assessment-estimated IR (HOMA-IR) value was calculated as HOMA-IR = [fasting insulin (mU/L) × fasting glucose (mmol/L)]/22.5 [23]. The IS index was determined as (1/[fasting insulin (mU/L) × fasting glucose (mmol/L)]) × 1000. The IR and IS models were constructed using fasting values of plasma insulin and glucose levels by using the HOMA approach, which has been validated against clamp measurements [33,43].

### 2.7. Western Blotting

The mice were killed at the end of the experiment. Their livers and gastrocnemius muscles were rapidly removed, minced roughly, and homogenized immediately. Western blotting was conducted using the approach detailed in another study [44]. We used Akt, phospho-Akt (Ser473), actin, GLUT4, and adiponectin antibodies procured from Cell Signaling Technology (Beverly, MA, USA). Sigma-Aldrich supplied antibodies against patatin-like phospholipid domain containing protein 3 (PNPLA3) and fatty acid synthase (FASN). Enhanced chemiluminescence reagents (Thermo Scientific, Rockford, MA, USA) were employed to obtain immunoreactive signals, which were detected using UVP ChemStudio (Analytik Jena, Upland, CA, USA). After that, Scion Image (Scion, Frederick, MD, USA) from the National Institute of Health was employed to measure protein expression and phosphorylation.

### 2.8. RNA Extraction and Real-Time Quantitative Polymerase Chain Reaction (PCR)

After the mice were sacrificed, total RNA was extracted from their eyeballs and liver tissues using TRIzol reagent (Sigma-Aldrich) per the manufacturer’s protocol. We then examined RNA concentration based on absorbance levels of 260 to 280 nm and 230 to 260 nm with a Qubit fluorometer (Invitrogen, Carlsbad, CA, USA). Subsequently, the RNA (1 μg) was subjected to reverse transcription into cDNA by using an iScript cDNA synthesis kit (Bio-Rad, Hercules, CA, USA) following the producer’s procedure. Subsequently, real-time polymerase chain reaction (PCR) was performed per the specifications of Bio-Rad’s iTaq Universal SYBR Green Supermix kit and with the application of the Bio-Rad CFX Connect Real-Time PCR system. In brief, the cycling conditions were as follows: 95 °C for 5 min, 45 cycles at 95 °C for 15 s, and 60 °C for 25 s. The expression level of every target gene was determined relative to *β-actin* levels, with these levels expressed in the 2^−ΔΔCt^ manner. The primers used for RT-qPCR are listed in Table 1.

### 2.9. Analysis of Chromium Content

After all experiments were complete, blood, urine, and several types of tissue (kidney, blood, liver, bone, fat, and muscle) were collected. Concentrations of chromium in samples were measured according to a prior report [40]. In brief, each sample (0.1 g of tissue and 25 µL of blood and urine) was digested in 65% nitric acid, and then each sample was subjected to digestion with nitric acid overnight (temperature: 100 °C). Chromium concentrations were determined using an ICP Mass Spectrometer (NexION 350X, PerkinElmer, MA, USA). Next, distilled water was used to dilute the digested solution to a 5 mL solution prior to measurements. The relative chromium recovery rate was calculated at 10 ppb and 100 ppb of the quantification levels by 5% (*n* = 5) and 8% (*n* = 5), respectively. The absorption data were plotted onto a 1–500 ppb standard curve, and regression analysis was performed to identify the total chromium level in samples (R^2^ > 0.996).

### 2.10. Measurement of Glutathione Peroxidase, Renal Catalase, and Superoxide Dismutase

We analyzed the antioxidant system’s functional activity to determine the enzymatic activity of glutathione peroxidase (GPx), hepatic catalase (CAT), and superoxide dismutase (SOD). Ice-cold saline (0.9% sodium chloride) was used to perfuse kidneys, after which they underwent homogenization in chilled potassium chloride (1.17%) by means of a homogenizer and in accordance with a previous description [22]. Next, the homogenates were gathered for analysis after 5 min of centrifugation at 800× *g* (4 °C). Finally, the supernatant was subjected to 20 min centrifugation at 10,500× *g* (also 4 °C) to obtain the postmitochondrial supernatant for the kidney samples to measure SOD, catalase, and GPx activity. Commercially available colorimetrical kits were used for these procedures (catalase: #K773-100, #GPx: K762-100, and SOD: #K335-100; BioVision) following manufacturer-recommended protocols. 

### 2.11. Statistical Analysis

Data are shown as the mean ± standard deviation. The *t*-test was employed for determining intergroup differences (*p* < 0.05 denoted a statistically significant difference). Additionally, we determined contingency data significance by means of Fisher’s exact test. 

## 3. Results

### 3.1. Imipramine Changed Food Efficiency, Morphometric Parameters, and Food Intake in Animals

Our preliminary investigation did not identify significantly different antiobesity effects in standard diet (SD)-fed animals that were and were not administered imipramine (Appendix A). Mice on an HFD were administered imipramine for 8 weeks to elevate their body weight, metabolic parameters, serum leptin levels, and leptin receptor levels (Figure 1). The body weight of the imipramine-treated mice at the end of the experiment was 21% higher (*p* < 0.01) than that of the HFD-fed controls (Figure 1a). Furthermore, weekly food consumption was 21% higher in mice after they were administered imipramine, and their increase was parallel with the food consumption increase in controls (Figure 1b). These changes took place alongside significantly increased daily food efficiency in mice administered imipramine; this efficiency was 20% higher than the control animals (Figure 1c). We also revealed that imipramine had a significant effect on serum leptin (Figure 1d) and leptin receptor levels (Figure 1e), with the treatment group exhibiting a 15% higher serum leptin level and an 11% lower leptin receptor level than the control group. Leptin also plays a role in food intake regulation [22].

### 3.2. Imipramine Increased Liver, Kidney, and Fat Pad Weights

Next, assessments were conducted to discover whether the indicated weight disparities were linked to body composition or adiposity alterations. Eight weeks after imipramine administration commenced, the body compositions of treated mice differed significantly from those of control mice in terms of their hearts, livers, kidneys, EWAT, and RWAT, but not in terms of their spleens (Figure 2). After 8 weeks of imipramine administration, the treated mice had significantly different body compositions from the control mice: their hearts, livers, kidneys, RWAT, and EWAT were 1.3, 1.3, 1.2, 1.2, and 1.3 times heavier than those of the control mice, respectively; moreover, when expressed as a proportion of body weight, these were 1.1, 1.2, 1.1, 1.2, and 1.1 times heavier than those of control mice, respectively. Spleen weights, as a proportion of the overall body weight, were not significantly different between the groups.

### 3.3. Imipramine Increased Hepatic Fat Accumulation and the Ratio of Large to Small Adipocytes but Reduced UCP1 mRNA Expression

A morphometric evaluation involving hematoxylin and eosin tissue staining for both groups revealed that the treated mice had considerably larger adipocytes in EWAT and RWAT and more liver fat compared with the controls (Figure 3a), indicating that imipramine promotes hepatic fat accumulation and may thus increase fat pad hypertrophy. Our analyses of alterations in fatty liver scores (Figure 3b) and RWAT (Figure 3c) and EWAT (Figure 3d) adipocyte size revealed significant differences between groups; the treatment mice had fatty liver scores almost 1.3 times higher than the control group’s scores. Additionally, the mean EWAT and RWAT adipocyte size was significantly larger (1.4 and 1.2 times, respectively) in the imipramine group, which agrees with the observation of higher fat pad weights in the mice administered imipramine. Moreover, we determined that imipramine had a significant influence on the mRNA expression of *UCP1*, with the treatment group exhibiting a 12% lower level than the controls (Figure 3e). In other words, the mice administered imipramine had a lower proportion of RWAT and EWAT adipocytes with a diameter of 0–40 and 40–80 μm (Table 2) but a higher proportion of those with a diameter of 80–120 and >120 μm. Therefore, although an increasing trend in RWAT and EWAT adipocyte size was discovered in response to mice being fed an HFD, imipramine expedited this size increase.

### 3.4. Imipramine Increased the Serum Levls of Serotonin, FGF21, ALT, and AST and the mRNA Levels of FABP4 and SREBP1

A report indicated that increased serotonin activity resulted in severe obesity and hepatic steatosis [45]. Our results indicate that the serum serotonin levels of imipramine-treated mice were 1.3 times higher than those of the controls (Figure 4a). In addition, the imipramine-treated group had 20% reduced FGF21 activity compared with that of the control group (Figure 4b). Serum ALT and AST were increased by 1.3 and 1.3 times, respectively, in the mice administered imipramine, indicating hepatic function (Figure 4c,d, respectively). FABP4 and SREBP1 are implicated in regulating many metabolic pathways, such as those related to type 2 diabetes, atherosclerosis, and hepatic lipid accumulation [38,43]. These genes reportedly have dominant effects in promoting FLD in rodents and humans [40]. In the current investigation, compared with control mice, mice administered imipramine had, respectively, 1.8 (Figure 4e) and 1.5 (Figure 4f) times higher liver mRNA levels of *FABP4* and *SREBP1*. The gene-associated activation of hepatic steatosis resulted in significantly increased AST and ALT levels; this was probably related to imipramine and its contribution to FLD, mediating the heightened mRNA expression of molecular mechanisms of hepatic lipid accumulation.

### 3.5. Imipramine Increased Serum Triglycerides and Hepatic Triglycerides, FASN Levels, and PNPLA3 Levels but Reduced Hepatic Adiponectin Levels

Levels of blood triglycerides (Figure 5a) and hepatic triglycerides (Figure 5b) were significantly higher in imipramine-treated mice than in control mice. Furthermore, we noted a 1.2-fold and 1.3-fold increase in blood triglyceride levels and hepatic triglycerides, respectively, compared to control mice. In addition, Western blot analysis performed on the liver of mice determined adiponectin, FASN, and PNPLA3 expression (Figure 5c). Adiponectin has mediated stimulation of energy expenditure and regulates the expression of hepatic genes critical for glucose and lipid metabolism [40]. Alternatively, FASN is a crucial liver enzyme for lipid homeostasis and triglyceride synthesis [46], and PNPLA3 is a marker that is pathologically characterized by the regulation of lipogenesis in obesity, nonalcoholic fatty liver disease, and cardiovascular disease [33]. Remarkably, FASN and PNPLA3 expressions in the liver of imipramine-treated mice were significantly higher (by 1.5-fold and 1.5-fold; Figure 5d,e, respectively) compared with those in the control mice. Moreover, adiponectin expressions in the imipramine-treated mice were significantly lower by 63% compared with those in the control mice (Figure 5f). These results indicate that imipramine increased the upregulation of lipogenesis in the liver and exacerbated HFD-induced fatty liver scores.

### 3.6. Imipramine Reduced Glucose Tolerance and Lowered Insulin Levels

To examine the effect of imipramine on glucose homeostasis, we conducted glucose tolerance tests. Significant imipramine-induced impairment of glucose tolerance was discovered in the imipramine group compared with the control group, as demonstrated by the IPGTT data presented in Figure 6a. Notably, imipramine administration resulted in significantly heightened fasting blood glucose. Moreover, imipramine administration resulted in a significant increase in fasting blood glucose levels at all time points after glucose supplementation through injection; in addition, after 120 min, blood glucose levels in the imipramine group were 1.1 times higher after glucose injection than they were before it. Furthermore, when comparing the control and imipramine groups, the AUC for glucose levels at 120 min was 1.1 times higher in the mice administered imipramine; this difference reached statistical significance (Figure 6b). With glucose intolerance defined as a >10 mmol L^−1^ blood glucose level at 120 min after an intraperitoneal glucose injection, we revealed that a large proportion of mice administered imipramine had glucose intolerance (Figure 6c) [39]. Imipramine-treated mice had significantly lower levels of serum insulin (Figure 6d). Therefore, HFD-fed mice that received imipramine had exacerbated diabetes symptoms, with more severe hyperglycemia and impaired glucose tolerance associated with hypoinsulinemia.

### 3.7. Imipramine Reduced IS through Changing the Expression of Phosphorylated Akt and GLUT4

Long-term imipramine treatment increased IR and diminished insulin insensitivity in HFD-fed mice. This finding is backed up by a significant difference in the respective HOMA-IR value and IS index (Figure 7a,b, respectively). Moreover, compared with the control group, the imipramine group had a significantly (1.2 times) higher HOMA-IR (Figure 7a). The IS index of the imipramine group was significantly lower (by 17%) (Figure 7b). Furthermore, we made an attempt to understand the mechanisms underlying glucose homeostasis in the mice fed an HFD through examining Akt phosphorylation and GLUT4 expression within muscles after imipramine treatment (Figure 7c). Significantly lower (11% and 25%, respectively) muscle Akt activation and GLUT4 expression were observed in the imipramine group (Figure 7d,e).

### 3.8. Imipramine Affected Chromium Levels in Tissues and Organs and Increased Chromium Urinary Loss 

Chromium (III) has important functions in glucose homeostasis, makes a major contribution to insulin action, and reduces IR [15,38]. Therefore, to evaluate whether imipramine-treated mice with HFD-induced obesity and glucose intolerance had altered chromium levels, we measured the chromium concentration in various harvested tissues and organs; the results are presented in Table 3. Chromium intake in the imipramine-treated mice was significantly higher (by 1.2 times) than that in the control mice. This finding was linked to the imipramine-treated mice being more hyperphagic than the control mice. Blood, bone, liver, muscle, and fat pad chromium levels were (significantly) 60%, 53%, 20%, 22%, and 24% lower, respectively, in the imipramine-treated mice than in the controls. By contrast, these chromium levels had an opposite trend in kidney and urine samples; average increases of 1.5 times and 2.2 times, respectively, were noted in the imipramine-treated mice compared with the controls, reaching the level of statistical significance.

### 3.9. Imipramine Induced Renal Injury, Increased Serum BUN and Creatinine, and Reduced Antioxidant Enzymes in the Kidneys

We next evaluated whether imipramine induced renal damage because evidence previously indicated that obesity, hyperlipidemia, and diabetes resulted in kidney injury [39]. Hematoxylin and eosin staining revealed that imipramine administration induced glomerulonephritis and inflammatory cell infiltration (Figure 8a). In addition, we observed significantly higher serum BUN and creatinine levels (1.2 times higher each) in the imipramine group (Figure 8b,c). Renal injury is linked to a reduction in antioxidant enzymes in the kidney; increased oxidant enzyme activity may result in alleviation of renal nephropathy and improve renal function [47,48]. Mice administered imipramine had significantly lower activities of antioxidant biomarkers CAT (Figure 8d), GPx (Figure 8e), and SOD (Figure 8f; 27%, 10%, and 13% lower, respectively).

### 3.10. Imipramine Aggravates Retina Damage and Increases the Gene Expression of iNOS, NF-κB, and COX-2 but Reduces IκBα Expression in the Eyeballs

Diabetes heightens oxidative stress, and such a heightened stress state may play a major role in diabetes complications, including retinal injury [47,48]. In hematoxylin and eosin-stained sections of retinas, the inner plexiform and nuclear layers (IPL and INL, respectively) of imipramine-treated mice were thinner than those of the control mice, and the ganglion cell layer was almost absent in the treated mice (Figure 9a). Ocular inflammation and associated complications are key causes of vision loss. In recent years, evidence has suggested that inflammation has a prominent role in the pathogenesis of several retinal conditions, such as DR [48]. Our results show that the gene expression of COX-2, iNOS, and NF-κB was higher in imipramine-treated mice compared with controls (1.5, 1.8, and 1.2 times higher, respectively; Figure 9b,c, respectively). By contrast, the gene expression of IκBα in imipramine-treated mice was 18% lower than that in controls (Figure 9d). These results indicate that imipramine increased the risk of DR by elevating inflammation and glucose uptake.

## 4. Discussion

We conducted this study to explore the impact of imipramine on obesity development in mice fed an HFD. Mice administered imipramine for 8 weeks had a higher obesity risk, more visceral fat, a higher fatty liver score, more severe glucose intolerance, increased IR, more severe kidney damage, and a larger AUC value 120 min after glucose was injected. Daily food efficiency, liver, kidney, and fat pad weight, fatty liver score, adipocyte size, serum and hepatic triglyceride levels, and serum BUN and creatinine also increased; however, reductions were noted in renal catalase, GPx, and SOD activities. Our results demonstrate that imipramine can increase weight gain and food intake and attenuate glucose homeostasis concomitant with hypoinsulinemia. Imipramine increases fatty liver scores and is linked to adipogenesis in the liver, including *FABP4* and *SREBP1* mRNA levels, in addition to adiponectin, FASN, and PNPLA3 activation. IR is typically associated with lower insulin signaling protein activity; our imipramine-treated hyperglycemic mice exhibited exacerbated glucose homeostasis induced by a reduction in phosphorylated Akt and GLUT4. Compared with the controls and after 56 days of imipramine treatment, the treated mice had significantly lower chromium levels in their bones, blood, livers, muscles, and fat pads; however, they had significantly higher levels in their urine and kidneys. Imipramine treatment also rendered mouse retinas abnormal and increased the expression of inflammatory-related genes, including COX-2, iNOS, and NF-κB, in the eyeballs. Finally, the treated mice also had increased obesity, hepatic fat accumulation, serum triglyceride levels, serum liver function indexes, and urinary chromium loss, and they exhibited insulin resistance, heightened glucose intolerance, renal injury, diabetic retinopathy, and thin IPLs and INLs corresponding to the development of hyperglycemia-associated fatty liver.

After mice were administered an HFD for 8 weeks, the diet made a contribution to obesity development. Weight gain increases occur when fat pad mass increases because of an increase in the number of new adipocytes from precursor cells or due to an enlarged adipocyte size because of fat storage [14,38]. Body weight gain is related to increased fat cell differentiation or fat pad weight caused by fat cell hypertrophy [14,39]. For this reason, the body weight of the imipramine-treated mice was greater than that of the control mice, and the weights of organs and tissue, including the kidneys, the liver, RWAT, and EWAT, were also greater. We suggest that the increases in body fat as well as RWAT, EWAT, and liver weights in the imipramine group may have been caused by increases in adipocyte size and fatty liver scores. Additionally, we obtained evidence that the increase in body fat may have been a result of lower energy expenditure, indicated by the mRNA expression in brown adipose tissue of UCP1. UCP1 expression occurs in the inner membrane of brown adipocyte mitochondria, creating heat through the uncoupling of oxidative phosphorylation, and thermogenesis induction is controlled by adrenaline. Hypothalamus feeding centers were linked to this thermogenic system for body temperature control, which enables the regulation of body weight and may well represent an efficient dual function related to the thermogenesis of brown fat [49]. Furthermore, leptin has roles in the hypothalamus regarding regulating food intake, and leptin synthesis occurs primarily within white adipose tissue [50]. The imipramine-treated mice ate more, and their serum leptin was higher than that of the controls. Leptin mediates food intake by means of sympathetic outflow to the hypothalamus, and leptin directly affects tissues through enhancement of lipid oxidation and a reduction in lipogenesis [51]. Imipramine therapy appeared to inhibit the leptin signaling pathway. Moreover, we found a decrease in leptin receptor levels in the imipramine-treated mice; the leptin receptor is a phenome of resistance to leptin. The finding that long-term use of imipramine may accelerate obesity through leptin signaling inhibition is valuable.

The accumulation of fat in the liver induces FLD, which is strongly associated with obesity [33,38]. The imipramine group had higher fatty liver scores (revealed through histopathology) and higher levels of the liver enzymes AST and ALT, which act as hepatic injury markers. Hepatocellular permeability is increased when liver injury is induced, and consequently, AST and ALT are released from intracellular spaces into plasma [52]. In addition, we analyzed *SREBP1* and *FABP4* mRNA expressions and discovered them to be associated with the expression of genes participating in lipid storage, hepatic steatosis, hepatic lipogenesis, and NAFLD pathogenesis [33,53]. Higher expression of SREBP1 and FABP4 mRNA in the liver was observed after long-term imipramine administration and caused liver injury owing to marked hepatic lipid infiltration. In addition, the activity of COX-2 in the obese imipramine-treated mice was significantly higher than in the control mice (Appendix A), which eventually led to inflammation and liver damage [54]. Serum AST and ALT levels were subsequently analyzed. Similar studies have found liver test abnormalities in patients administered tricyclic antidepressants, such as aminotransferase abnormalities and cholestatic hepatitis [55]. A report indicated that increased systemic serotonin activity through the knockout of the serotonin transporter resulted in IR, severe obesity, and hepatic steatosis [53]. Our findings corroborate those of other studies related to imipramine-treated HFD-fed mice; these mice exhibited substantially increased fat pad and liver weights, serum and hepatic triglyceride levels, and fatty liver scores in response to increased serum serotonin levels. For safety reasons, when imipramine use is linked to higher risks in individuals with depression and underlying liver disease, it is recommended that serum AST and ALT be regularly monitored during clinical trials despite the fact that no such recommendation has been formally made for related investigations.

Obesity is highly associated with elevated triglycerides [56]. In addition, the hallmark of NAFLD is accumulated triglycerides in hepatocytes’ cytoplasm [57]. We explored serum and hepatic triglyceride levels. Our imipramine group exhibited elevated serum and hepatic triglyceride levels compared with the control group. This may have been related to imipramine decreasing fibroblast growth factor-21 activity, which has major effects on metabolic parameters, including on lipid and glucose homeostasis, and which promotes fast weight loss [58]. The finding may be linked to lower FGF21 levels preventing a reduction in excess lipid levels in hepatocytes; this may result in lipid overload-related stress and activate the release of several proinflammatory factors (e.g., tumor necrosis factor-α (TNF-α)) [59,60]. FASN may have a role in body weight regulation and the development of obesity [61]. Research has revealed a significant correlation of FASN expression with the degree of steatosis in primary human hepatocytes in vitro as well as in experimental murine models and in the livers of patients with NAFLD in vivo [62]. Furthermore, hepatic inflammation may be promoted by PNPLA3 during NAFLD progression through an increase in TNF-α expression and activation of endoplasmic reticulum stress-mediated and NF-kB-related inflammation in NAFLD [63]. In addition, adiponectin has antidiabetic characteristics owing to its insulin-mimetic and insulin-sensitizing effects, and furthermore, anti-inflammatory and anti-atherosclerotic effects have been consistently reported [64]. As expected, in this study, adiponectin was reduced in the imipramine-treated mice. Taken together, the results indicate that imipramine induced a marked intrahepatic accumulation of lipids, and this exacerbated liver damage in mice with HFD-induced NAFLD. The activation of multiple inflammatory signaling pathways by imipramine may accelerate the development of liver damage. Therefore, regulation of imipramine in these pathways may indicate novel treatment targets for these increasingly less common sequelae of hepatic side effects after imipramine administration.

Reports indicated that tricyclic antidepressants elevate and reduce glucose and insulin levels, respectively, in humans and animals, contributing to greater metabolic risks in nondiabetic patients with depression or the deterioration of glucose metabolism in individuals with depression and type 2 diabetes through the aggravation of glucose intolerance [65,66,67]. Moreover, evidence indicates that TCAs, such as doxepin and amitriptyline, induce hyperglycemia by inhibiting insulin release from the pancreas and inhibit glucose transport, resulting in decreased glucose uptake [68,69]. Our results indicate that imipramine exacerbates diabetes mellitus complications by reducing serum insulin levels, as evidenced by the fact that chronic imipramine use resulted in a reduced proportion of β-cells in mouse pancreas sections (Appendix A). Generally, a reduction in Akt phosphorylation and serum FGF21 level leads to the attenuation of insulin signaling, which impairs glucose homeostasis and enhances IR [33,39]. GLUT4 facilitates insulin-promoted glucose uptake into adipose tissue and muscle, and reduced GLUT4 expression levels lower insulin-mediated glucose uptake; exacerbated hyperglycemia may be the mechanism of action, as indicated by heightened glucose intolerance on the basis of an IPGTT [70]. Considerable basal glucose transport reductions suggest strong IR and glucose intolerance in mice that have diabetes and a selective deficiency of GLUT4 in their muscles [33]. Collectively, these findings indicate that HFD-fed mice administered imipramine have severe hyperglycemia and IS suppression.

Chromium administration may be a crucial adjunct therapy for type 2 diabetes because it may be involved in glucose metabolism through potentiating insulin’s actions [38]. This action of chromium has a role in metabolism, and a consequence of this action is glucose metabolism alteration. Our results also indicate that chromium levels in the muscles, blood, bones, liver, and fat pads of HFD-fed mice administered imipramine were significantly lower than in control mice. These results show that imipramine negatively affects chromium accumulation in tissues [39]. Similarly, serum chromium levels were significantly decreased in patients with uncontrolled type 2 diabetes [71]. Therefore, prolonged imipramine intake could change chromium concentrations in various organs and tissues. Chromium mobilization or redistribution prompted the worsening of the hyperglycemic status of our mice administered imipramine.

Large amounts of trace elements are excreted in urine; however, the majority of them are absorbed again at the proximal renal tubule [34]. Although chromium levels were low in the metabolic tissue and blood of our imipramine group, they were high in urine and kidney samples. We surmised that the imipramine resulted in chromium release from metabolic tissues into the kidneys, thus increasing urinary chromium loss. Greater urine excretion and lower trace metal reabsorption are generally kidney damage responses [15]. This supposition is backed up by the higher renal function indexes (creatine and BUN) in our mice administered imipramine. Higher creatine and BUN concentrations are known to be kidney disease progression risk factors in those with moderate or severe CKD [72]. Additionally, imipramine caused lesions in mice and glomerulonephritis. Collectively, the results demonstrate that chronic intake of imipramine causes kidney injury; inflammatory cell infiltration in the renal interstitium was observed alongside increased biomarkers of nephropathy, namely, creatine and BUN. Detoxicant and antioxidant enzymes provide crucial protection to the kidney [73]. Due to their transport-related functions, the kidneys have an active oxidative metabolism, and this action causes ROS production [74]. We detected lowered GPx, catalase, and SOD activities in the treated mice compared with our controls. Furthermore, immunohistochemical (IHC) staining demonstrated that mice administered imipramine had heightened levels of renal inflammatory cytokines, including TNF-α (Appendix A). The results indicate that chronic and continuous imipramine administration can lead to hyperglycemia-induced kidney damage and long-term inflammation, exacerbating injuries caused by reduced antioxidant levels and oxidative stress. Researchers should investigate whether drugs other than imipramine can be used as replacement or adjunct therapies to alleviate kidney-related side effects in diabetes patients with depression.

Hyperglycemia is a common diabetes symptom, and it may induce oxidative stress and inflammatory injuries during DR development and later result in vascular dysfunctions [75]. DR can lead to blindness [76]. IR and chronic hyperglycemia are key factors related to blood–retina barrier dysfunction and retinal damage related to diabetes [77]. Various neuronal cells, including amacrine and ganglion cells, are affected by pathogenic changes that are manifestations of morphologic abnormalities and neuronal apoptosis [78]. Numerous physiological and molecular abnormalities develop in the retinas of individuals with diabetes, and the manifestations seem to be related to inflammation [79]. Upregulated COX-2 and iNOS have been noted in rodent and human retinas [80]. Moreover, NF-κB, which is inhibited by IκBα, is a widely expressed inducible transcription factor known to be a crucial regulator of various genes with roles in immune and inflammatory responses and cellular apoptosis and proliferation. NF-κB activation is induced by proinflammatory proteins such as iNOS [81]. Diabetes activates NF-κB in the retinas of rodents [82]. Through hematoxylin and eosin staining, we discovered that retina cells in the mice administered imipramine had more severe injury than such cells in control mice; RT-PCR revealed that the gene expression of inflammatory markers (e.g., iNOS, COX-2, and NF-κB) was heightened, and IκBα expression was reduced in mice administered imipramine.

Our findings suggest that an HFD can be used to create a mouse model of both hyperglycemia and obesity, and that prolonged imipramine intake in mice causes weight gain; increased food intake; increased kidney, liver, and fat pad weight; heightened serum and hepatic triglyceride, ALT, and AST levels; and increased fatty liver scores and adipocyte sizes. These findings could, indeed, be corroborated by lipid accumulation and lipogenesis activation for the regulation of nutrient metabolism. Furthermore, administering mice imipramine worsened their hyperglycemia, expedited glucose intolerance with glomerulonephritis, and caused retinopathy. The reduced expression of Akt and GLUT4 in skeletal muscles was possibly a factor in the indicated reduction in glucose metabolism. Imipramine altered the distribution of chromium in tissues and organs and increased the urinary loss of chromium. These findings point to the attenuation of glucose homeostasis and IS and the enhancement of IR. Moreover, the HFD slowed glucose homeostasis in these mice, but imipramine expedited the manifestation of diabetes symptoms. Thus, imipramine could have side effects related to kidney damage, FLD, and diabetes (Appendix A). Chronic imipramine intake could exacerbate conditions such as obesity, diabetes, CKD, and DR.

## 5. Conclusions

The present study reveals new information indicating that constant intake of imipramine in mice as part of an HFD had marked effects in terms of obesity and hyperglycemia inducement. Parallel alterations in weight gain (including fat pad and body weight), food efficiency, serum triglyceride levels, serum AST and ALT levels, fatty liver scores, adipocyte size, and glucose intolerance were indicated. The current investigation provides evidence of the mechanisms behind the imipramine-induced exacerbation of hyperglycemia through decreasing GLUT4 expression. Nephropathy in obese mice administered imipramine may have been due to the exacerbation of hyperglycemia, and reductions in antioxidant enzymes may have worsened nephropathy. DR as well as thinner IPLs and INLs among mice administered imipramine was noted, and the mechanisms behind these manifestations were the increased expression levels of inflammatory COX-2, iNOS, and NF-κB. Blood glucose alterations, kidney and liver function, and eyesight changes must be carefully assessed following initial imipramine intake to ensure early recognition of these rare side effects, especially in patients and animals receiving antidepressants. Such recognition would allow for early identification and treatment to prevent the development of DR, CKD, obesity, and hyperglycemia.

## Figures and Tables

**Figure 1 vetsci-08-00189-f001:**
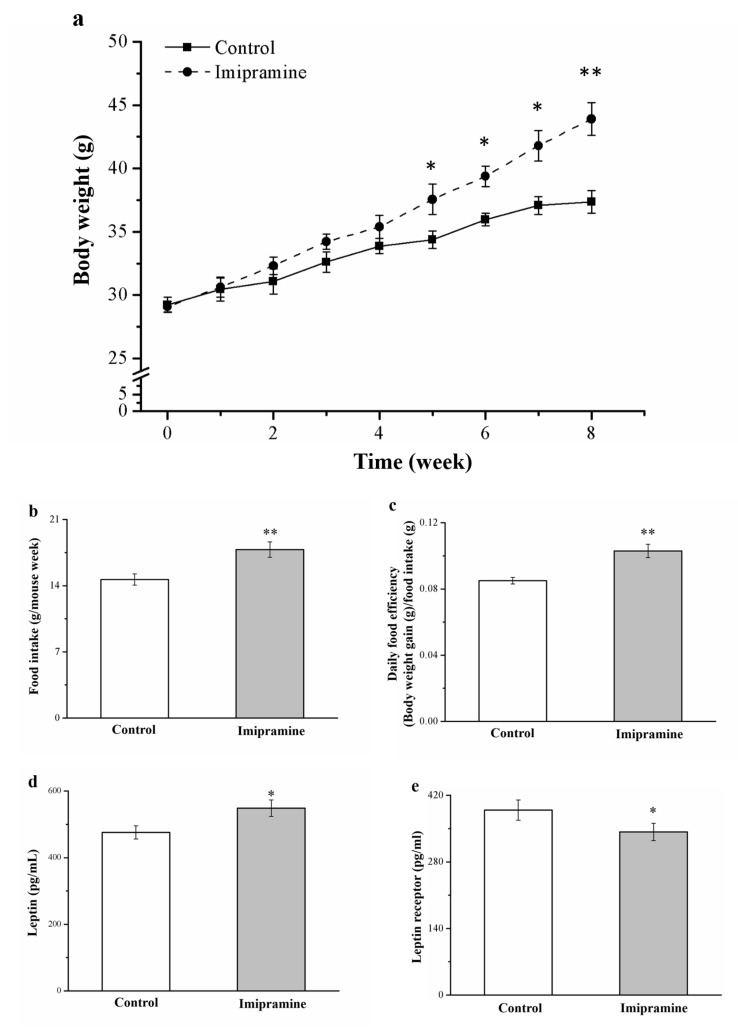
Effects of imipramine on (**a**) weekly body weight changes and (**b**) food intake in each group (measured weekly). (**c**) Daily food efficiency, (**d**) serum leptin levels, and (**e**) leptin receptor levels were measured in HFD-fed controls and imipramine-treated obese mice. For both groups, data are in the form of mean ± standard deviation (*n* = 8). * *p* < 0.05; ** *p* < 0.01.

**Figure 2 vetsci-08-00189-f002:**
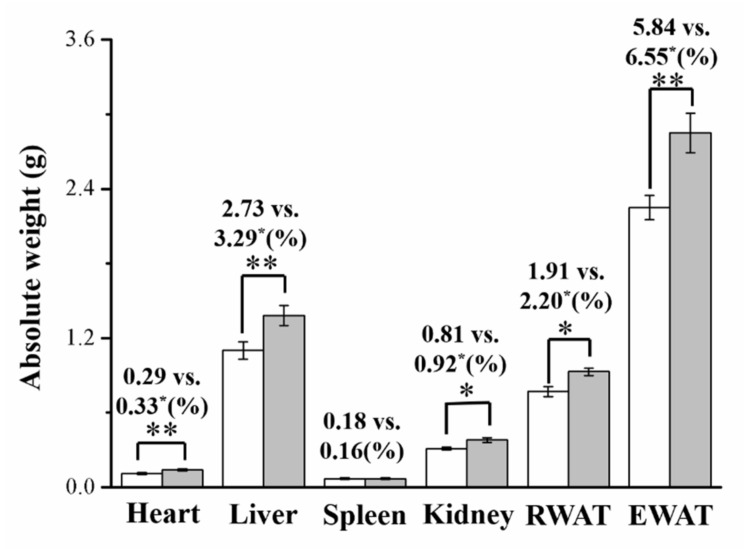
Effects of imipramine on total heart, retroperitoneal white adipose tissue (RWAT), spleen, liver, kidney, and epididymal white adipose tissue (EWAT) weights (shown as a percentage of total body weight). Both groups’ data are in the form of mean ± standard deviation (*n* = 8). * *p* < 0.05; ** *p* < 0.01.

**Figure 3 vetsci-08-00189-f003:**
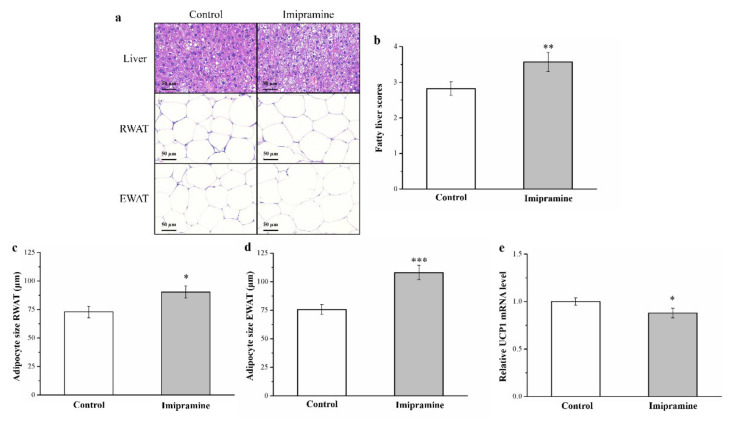
Effects of imipramine on (**a**) the mouse livers, retroperitoneal white adipose tissue (RWAT), and epididymal white adipose tissue (EWAT) of control and treated mice, as determined through hematoxylin and eosin staining (magnification, 200×). (**b**) Fatty liver score findings. (**c**) RWAT and (**d**) EWAT adipocyte sizes and (**e**) the mRNA expression of *UCP1* in brown adipose tissue were measured in HFD-fed control and imipramine-treated obese mice. Data are in the form of mean ± standard deviation (*n* = 8). * *p* < 0.05; ** *p* < 0.01; *** *p* < 0.001.

**Figure 4 vetsci-08-00189-f004:**
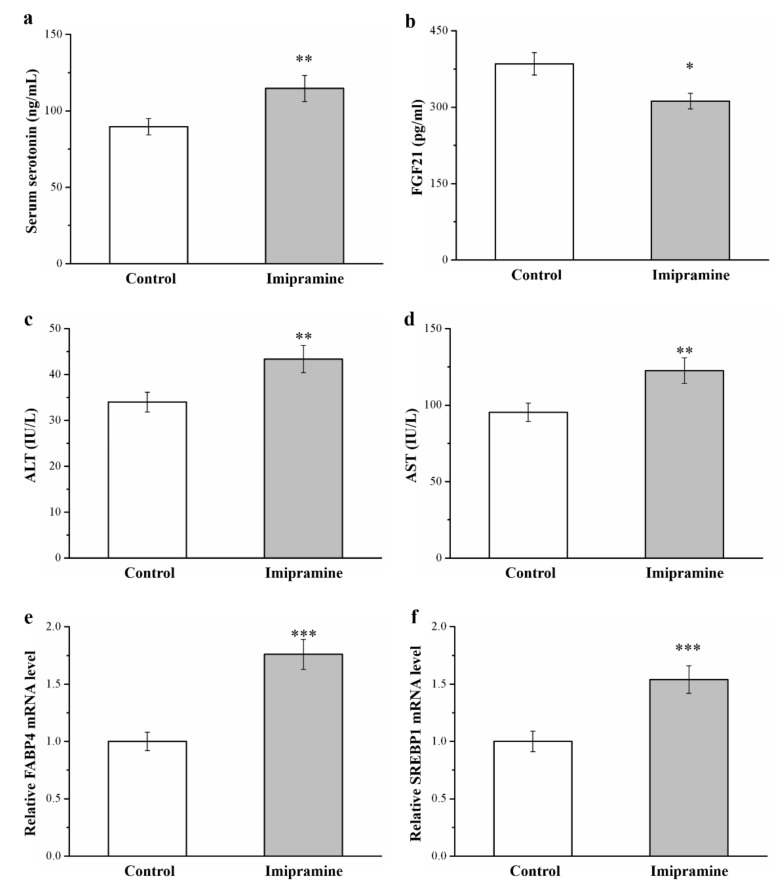
Effects of imipramine on (**a**) serum serotonin levels, (**b**) serum fibroblast growth factor-21 (FGF21) levels, (**c**) serum alanine aminotransferase (ALT) levels, (**d**) serum aspartate aminotransferase (AST) levels, and hepatic mRNA levels of (**e**) FABP4 and (**f**) SREBP1 measured in HFD-fed control and obese imipramine-treated mice. Data are in the form of mean ± standard deviation (*n* = 8). * *p* < 0.05; ** *p* < 0.01; *** *p* < 0.001.

**Figure 5 vetsci-08-00189-f005:**
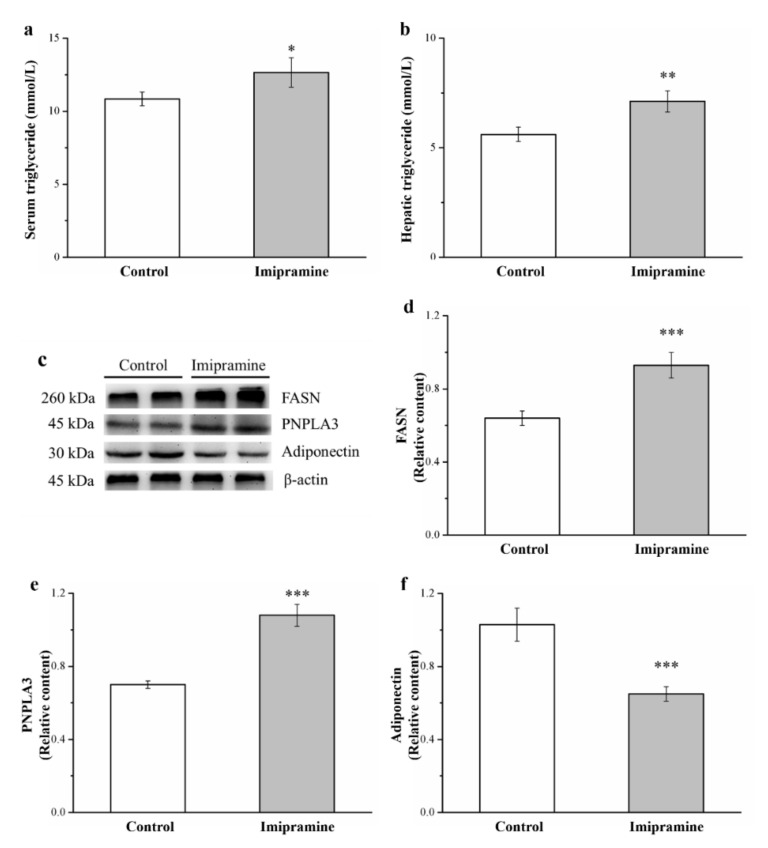
Effects of imipramine on (**a**) serum triglycerides and (**b**) hepatic triglycerides. (**c**) A representative image showing blots of liver extracts. The hepatic expression levels of (**d**) fatty acid synthase (FASN), (**e**) patatin-like phospholipid domain containing protein 3 (PNPLA3), and (**f**) adiponectin were measured. Data are in the form of mean ± standard deviation (*n* = 8). * *p* < 0.05; ** *p* < 0.01; *** *p* < 0.001.

**Figure 6 vetsci-08-00189-f006:**
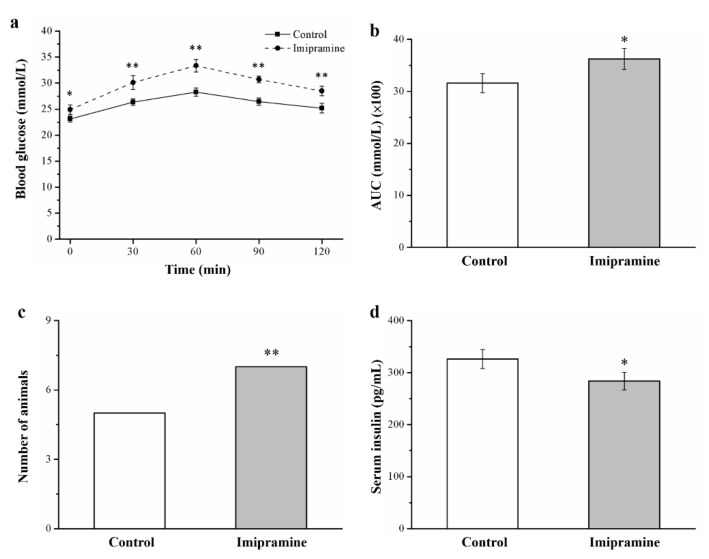
Impact of imipramine on (**a**) blood glucose tolerance score. (**b**) Area under the curve (AUC) at 120 min post-glucose injection of blood glucose levels. (**c**) Glucose intolerance (Fisher’s exact test). (**d**) Serum insulin levels. Data are in the form of mean ± standard deviation (*n* = 8). * *p* < 0.05; ** *p* < 0.01.

**Figure 7 vetsci-08-00189-f007:**
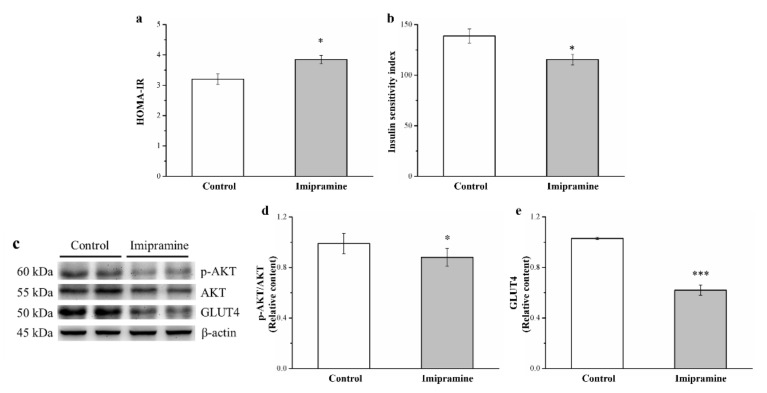
Effects of imipramine on the (**a**) homeostasis model assessment-estimated IR (HOMA-IR) index and (**b**) insulin sensitivity (IS) index. (**c**) Representative blot of gastrocnemius muscle extracts. Expression levels of (**d**) phosphorylated Akt and (**e**) GLUT4. Data are in the form of mean ± standard deviation (*n* = 8). * *p* < 0.05; *** *p* < 0.001.

**Figure 8 vetsci-08-00189-f008:**
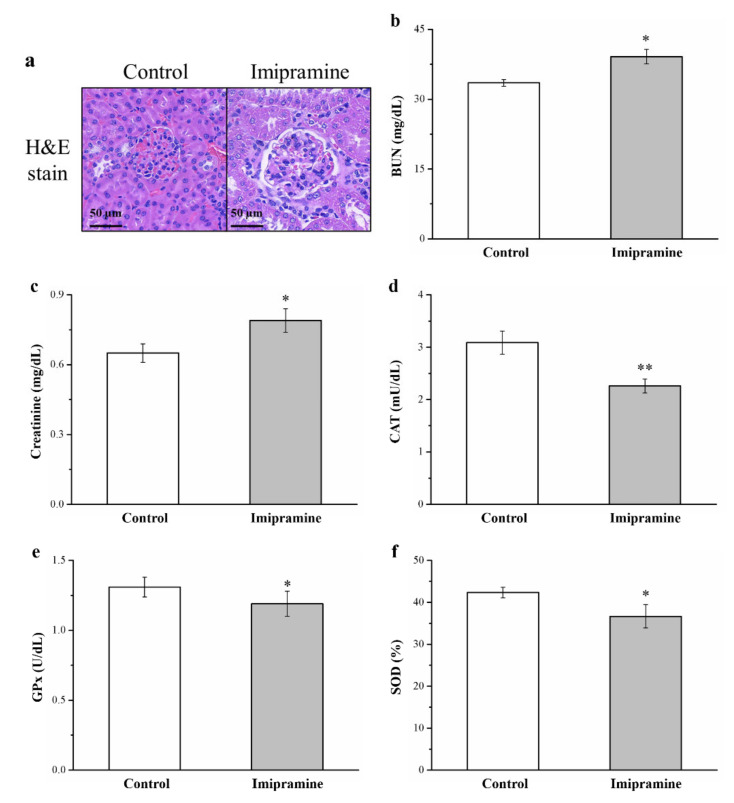
Effect of imipramine on (**a**) glomerular morphologies according to hematoxylin and eosin staining (magnification, 200×). (**b**) Serum BUN, (**c**) serum creatinine, (**d**) renal catalase (CAT) activity, (**e**) renal glutathione peroxidase (GPx) activity, and (**f**) renal superoxide dismutase (SOD) activity. Data are in the form of mean ± standard deviation (*n* = 8). * *p* < 0.05; ** *p* < 0.01.

**Figure 9 vetsci-08-00189-f009:**
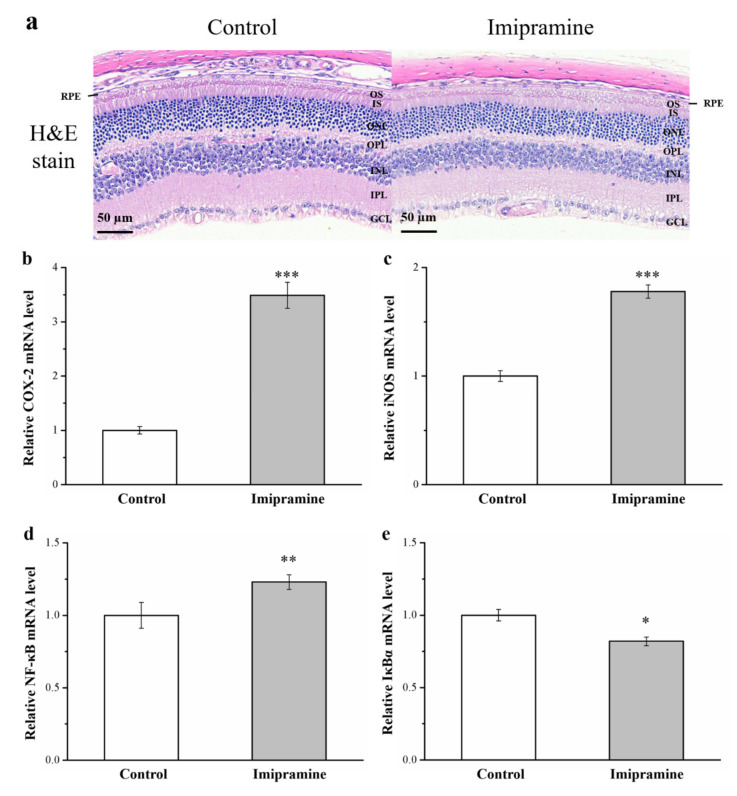
Effects of imipramine on (**a**) retinal morphology according to hematoxylin and eosin staining (magnification, 200×). (**b**) Cyclooxygenase-2 (COX-2), (**c**) inducible nitric oxide synthase (iNOS), (**d**) nuclear factor κB (NF-κB), and (**e**) inhibitory κBα (IκBα) mRNA levels. Data are in the form of mean ± standard deviation (*n* = 8). GCL, ganglion cell layer; IPL, inner plexiform layer; INL, inner nuclear layer; OPL, outer plexiform layer; ONL, outer nuclear layer; IS/OS, inner/outer segment junction; RPE, retinal pigment epithelium. * *p* < 0.05; ** *p* < 0.01; *** *p* < 0.001.

**Table 1 vetsci-08-00189-t001:** Sequences of primers used for RT-qPCR.

Gene	Primer Sequences
Fatty acid-binding protein 4 (FABP4)	Forward: 5′-GATGAAATCACCGCAGACGACA-3′ Reverse: 5′-ATTGTGGTCGACTTTCCATCCC-3′
Sterol regulatory element–binding protein 1 (SREBP1)	Forward: 5′-CGGAAGCTGTCGGGGTAG-3′ Reverse: 5′-GTTGTTGATGAGCTGGAGCA-3′
Inducible nitric oxide synthase (iNOS)	Forward: 5′-CCTCCTCCACCCTACCAAGT-3′ Rreverse: 5′-CACCCAAAGTGCTTCAGTCA-3′
Cyclooxygenase-2 (COX-2)	Forward: 5′-TTCAAAAGAAGTGCTGGAAAAGGTTCT-3′ Rreverse: 5′-AGATCATCTCTACCTGAGTGTCCTT-3′
Nuclear factor κB (NF-κB)	Forward: 5′-GCAACTCTGTCCTGCACCTA-3′ Reverse: 5′-CTGCTCCTGAGCGTTGACTT-3′
Inhibitory κBα (IκBα)	Forward: 5′-AAGTGATCCGCCAGGTGAAG-3′ Reverse: 5′- CTGCTCACAGGCAAGGTGTA -3
Uncoupling protein 1 (UCP1)	Forward: 5′-GGCCTCTACGACTCAGTCCA-3′ Reverse: 5′-TAAGCCGGCTGAGATCTTGT-3′
β-actin	Forward: 5′-GGCTGTATTCCCCTCCATCG-3′ Reverse: 5′-CCAGTTGGTAACAATGCCATGT-3′

**Table 2 vetsci-08-00189-t002:** Influence of imipramine on fat cell sizes in HFD-fed controls and imipramine-treated mice.

Variable	Control	Imipramine
Retroperitoneal white adipose tissue (RWAT)		
Adipocyte diameter		
0–40 μm (%)	17.59 ± 0.43	5.78 ± 0.04 ***
40–80 μm (%)	58.97 ± 4.31	30.98 ± 1.93 ***
80–120 μm (%)	21.44 ± 0.62	57.33 ± 3.86 ***
>120 μm (%)	0 ± 0	5.91 ± 0.77 ***
Epididymal white adipose tissue (EWAT)		
Adipocyte diameter		
0–40 μm (%)	5.62 ± 0.29	0 ± 0 ***
40–80 μm (%)	71.37 ± 5.82	20.47 ± 1.06 ***
80–120 μm (%)	23.01 ± 1.3	67.09 ± 4.84 ***
>120 μm (%)	0 ± 0	12.44 ± 0.92 ***

Data are in the form of mean ± standard deviation (*n* = 8). *** *p* < 0.001.

**Table 3 vetsci-08-00189-t003:** Chromium concentrations in the organs and tissues of HFD-fed controls and imipramine-treated mice.

Variable	Control	Imipramine
Chromium intake/mouse/week (μg)	16.42 ± 0.66	19.96 ± 0.91 *
Blood (ng/mL)	182.28 ± 8.51	72.36 ± 5.78 ***
Bone (ng/g)	326.58 ± 9.26	152.27 ± 6.31 ***
Liver (ng/g)	76.27 ± 6.83	60.61 ± 4.25 **
Muscle (ng/g)	52.36 ± 4.62	40.84 ± 3.73 **
Epididymal fat pads (ng/g)	50.59 ± 4.61	38.52 ± 2.54 **
Kidney (ng/g)	100.51 ± 3.75	151.73 ± 6.29 ***
Urine (ng/mL)	51.18± 3.65	112.41 ± 4.23 ***

Data for the two groups are in the form of mean ± standard deviation (*n* = 8). * *p* < 0.05; ** *p* < 0.01; *** *p* < 0.001.

## Data Availability

The data presented in this study are available on request from the corresponding author.

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
