# Peer review of "Imipramine Accelerates Nonalcoholic Fatty Liver Disease, Renal Impairment, Diabetic Retinopathy, Insulin Resistance, and Urinary Chromium Loss in Obese Mice"

_vetsci, 2021, doi:10.3390/vetsci8090189_

Round 1
Reviewer 1 Report
Imipramine is a tricyclic antidepressant that has been approved for treating depression and anxiety in patients and animals and that has relatively mild side effects. In the present study, the mechanisms of imipramine-associated disruption to metabolism and negative hepatic, renal, and retinal effects are not well explored. However, the results are not rigorors and enough novel, and provide little new insight to the readers in the discipline. In my opinion, the manuscript has not meet the publication standard of Veterinary Sciences journal.
Reviewer 2 Report
The manuscript reports an impressive number of tests carried out by the authors to investigate a wide range of adverse effects induced by the use of imipramine, demonstrating an excellent expertise on the various techniques necessary to investigate the physiology and biochemistry of the examined phenomena; all tests were conducted with scientific rigor and accurately described, so as to allow other researchers to repeat / deepen the reported trials, and statistical analyzes are also correct.
The treatment of animals is ethically appropriate.
The iconography is well organized and understandable, as are the tables.
The bibliographic base is extensive and up to date.
The results obtained in the various tests well support the conclusions, which are extremely interesting considering the widespread use of TC antidepressants in both human and veterinary medicine.
Publication in the present form is recommended
Reviewer 3 Report
Reviewer comments and suggestions
The manuscript investigated the effect of imipramine’s on C57BL6/J mice that were exposed to a high fat diet (HFD) to study different risk factors such as obesity, fatty liver scores, glucose homeostasis, hepatic damage, distribution of chromium, and retinal/renal impairments.
The study reported that obese mice receiving imipramine treatment had a higher body, epididymal fat pad, and liver weights; a various markers of lipid and kidney compared with the controls also fed an HFD.
Furthermore, the obese mice that received imipramine treatment various abnormalities of glucose homeostasis parameters and increased chromium loss through urine.
In summary, the paper concluded that administration of imipramine may result in the exacerbation of nonalcoholic fatty liver disease, diabetes, diabetic retinopathy, and kidney injury.
Below are the comments to be incorporated into the manuscript
- Line 45-47, please check the correctness of the information. Here the author has used “ameliorated” which creates confusion.
- Line 81, the author cited recent studies here but the author used only one reference
- Line 128-133 please shorten this, actually, the meaning of all was similar to each other.
- Line 230, the sentence needs to modify, “at the conclusion of every experiment”
- RNA extraction and real-time quantitative polymerase chain reaction (PCR) the section need to modify as the representation was not good. The primers should not be in paragraph
- The legend in the paper should not be in short form, please check all the tables and figures
7.Figure 5 and 6 It would be better if the author explains here the source of samples
- Figure 6 © please check the y axis information
- It should be short and clear, repeating the same information is not a good idea
- A mechanistic figure was lacking in the paper
- Please check the references such as 18,38 and For figure S1 representation was not good
Author Response
Please see the attachment.

This manuscript is a resubmission of an earlier submission. The following is a list of the peer review reports and author responses from that submission.
Round 1
Reviewer 1 Report
- Figure S8, the picture title of a is wrong, not HE staining.
- The format of Reference is not uniform, such as 14, 70, etc.
- The description of enzymes should be activity, not level, refer to PMID: 32474337.
- For the NF-KB, COX-2, iNOS, et al., the authors only tested the mRNA expression, why not assay the protein levels? Refer to PMID: 33254389, PMID: 33047770.
- For the evaluation of fatty liver, oil red O staining is classical method, why did the authors only use HE staining rather than oil red O staining ? Refer to PMID: 31835194, PMID: 31918140.
- The English should be improved, including punctuation.
- The Conclusion section is redundancy, and need a concise presentation.
- The Discussion section lacks some strict logic among the mentioned topics.
Reviewer 2 Report
This study by Chang et al., describes the effects observed in mice treated with Imipramine consequent with high fat diet induced obesity. The observations quite conclusively demonstrate that chronic Imipramine treatment increases adiposity and glucose intolerance, as well as increasing hepatic, renal and retinal injury. Given the potential chronic use of this antidepressant in humans, the findings appear to be of high importance.
The manuscript could benefit from a more concise and focused discussion regarding the effects of Imipramine treatment, rather than a description of each measure or gene/protein. Can the authors speculate on whether Imipramine is effecting all these organs independently, or are the alterations in glucose metabolism, or chromium depletion, are driving the hepatic, renal and retinal injury. Furthermore, many interesting and relevant results were placed in supplementary figures and not mentioned in the results.
Major criticisms:
- The article should be edited for English language. While much of the article is well written, many word choices are misleading and in some cases indicate the opposite to what the results show. This is particularly evident in the summary where the first and last sentences indicate the opposite to what the results show.
- Remove non-scientific language such as heightening (P3, line 101).
- Refrain from describing things as ‘altered’ (P3 line 105), as it is then unclear whether they are improved or not.
- The supplementary figures, especially S3-S8, should be integrated into the main results section. In my opinion these are some of the most interesting observations and should not only be used to support discussion.
- It is unclear what you are referring to in P4, line 164-165. Are these body weights? If so, how is this significantly different since there is only 0.12 g difference.
- Please include a body weight curve including the initial body weight of the mice. The body weight gain does not appear to equate to the body mass differences stated, please explain.
- How is the model relevant to humans? How does the dose given to mice compare to prescribed doses in humans?
- P11, line 419-422 and Fig 6C: Where does this definition of glucose intolerance come from? Please provide references or additional data to support this claim.
- Reference 40 does not validate the use of HOMA-IR index or IS index in mice or compare this to clamp. Please provide appropriate reference to validate these indexes.
- Please ensure all references are relevant and appropriate.
Minor criticisms are as follows:
- How is food efficiency determined? Was the caloric content of feces included in the calculation?
- P8, section 3.: The descriptions of the heart appear contradictory.
- P8, line 344-348: Since there is greater lipid accumulation in all tissues, both liver and adipose, this sentence is not accurate.
- At what metabolic state were the mice when blood was collected for assessment of insulin levels. i.e. fed/fasting, if so how long?
- Fig 9A: Renal morphology also appears very different at the top of the image below the muscle layer. Please comment. Also, please annotate all layers of the retina.
- How does Imipramine effects the measured parameters in the absence of a HFD. Please comment.
- The y-axis are seemingly random intervals which is challenging for the reader to interpret. Please be consistent or use multiples of 5, 10, 50 or 100.
Round 2
Reviewer 1 Report
The authors have not well addressed the concerns, and the research is lack of innovation and the idea and methods are a little superficial. For the Discussion, it can not clarify the viewpoints very well with the results. Therefore, in my opinion, this manuscript is not acceptable for publication in Animals due to puzzling problems.